# An Engineered IFNγ-Antibody Fusion Protein with Improved Tumor-Homing Properties

**DOI:** 10.3390/pharmaceutics15020377

**Published:** 2023-01-22

**Authors:** Cesare Di Nitto, Ettore Gilardoni, Jacqueline Mock, Lisa Nadal, Tobias Weiss, Michael Weller, Frauke Seehusen, Chiara Libbra, Emanuele Puca, Dario Neri, Roberto De Luca

**Affiliations:** 1Philochem AG, Libernstrasse 3, 8112 Otelfingen, Switzerland; 2Department of Neurology and Clinical Neuroscience Center, University Hospital Zurich, University of Zurich, 8091 Zurich, Switzerland; 3Laboratory for Animal Model Pathology (LAMP), Institute of Veterinary Pathology, Vetsuisse Faculty, University of Zurich, 8057 Zurich, Switzerland; 4Philogen S.p.A., Piazza La Lizza 7, 53100 Siena, Italy

**Keywords:** interferon-gamma, antibody fusion proteins, protein engineering, cytokines, immunotherapy

## Abstract

Interferon-gamma (IFNγ) is one of the central cytokines produced by the innate and adaptive immune systems. IFNγ directly favors tumor growth control by enhancing the immunogenicity of tumor cells, induces IP-10 secretion facilitating (CXCR3+) immune cell infiltration, and can prime macrophages to an M1-like phenotype inducing proinflammatory cytokine release. We had previously reported that the targeted delivery of IFNγ to neoplastic lesions may be limited by the trapping of IFNγ-based products by cognate receptors found in different organs. Here we describe a novel fusion protein consisting of the L19 antibody, specific to the alternatively spliced extra-domain B of fibronectin (EDB), fused to a variant of IFNγ with reduced affinity to its cognate receptor. The product (named L19-IFNγ KRG) selectively localized to tumors in mice, showed favorable pharmacokinetic profiles in monkeys and regained biological activity upon antigen binding. The fusion protein was investigated in two murine models of cancer, both as monotherapy and in combination with therapeutic modalities which are frequently used for cancer therapy. L19-IFNγ KRG induced tumor growth retardation and increased the intratumoral concentration of T cells and NK cells in combination with anti-PD-1.

## 1. Introduction

The use of immunomodulatory molecules to enhance the activity of the immune system against cancer cells has steadily grown, leading to numerous novel biopharmaceuticals which have entered clinical trials [1,2,3,4]. Pro-inflammatory cytokines are an important class of immunostimulatory products that are being considered for immunotherapy applications, either as recombinant products or as building blocks for the production of fusion proteins [5]. Some recombinant cytokines have gained marketing authorization, including IL2, TNF, IFNα, IFNβ, IFNγ, GM-CSF, G-CSF, and IL11 [6,7]. Recombinant IL2 (Proleukin^®^), which is approved for the treatment of renal cell carcinoma and metastatic melanoma [8,9], induced durable complete responses in a small proportion of cancer patients [10]. However, recombinant cytokine products often cause substantial toxicity even at low doses, preventing escalation to therapeutically active regimens [11,12].

A number of strategies have been considered to increase the therapeutic index of cytokine products [13,14]. Several groups, including our own, have previously shown that the antibody-mediated targeted delivery of certain cytokine payloads to the tumor microenvironment may allow increasing the concentration of the cytokine at the site of disease while helping spare toxicity in healthy organs [15,16,17,18].

Interferon-gamma (IFNγ) is a pleiotropic cytokine released by innate and adaptive immune cells induced by interleukin-12 (IL-12) and interleukin-18 (IL-18) signaling cascades [19]. It has a heterogenous C-terminus with a predominant active human form of 138 amino acids in length [20]. Upon binding to its receptor (expressed on CD4+, CD8+T-cells and NK-cells), IFNγ can trigger biologically relevant anti-tumor activities. The cytokine can boost the expression of MHC class I molecules on tumor cells [21]. IFNγ mediates the recruitment of T-cells at the site of disease through secretion of IP-10 [22]. CXCL-10 or IP-10 is an 8.7 kDa protein constitutively expressed at low levels in secondary lymphoid organs [23]. High expression of this chemokine can be easily induced by IFNγ in a variety of cells including monocytes, neutrophils or activated T cells. This protein can act as a chemoattractant for circulating NK, CD4+ and CD8+ T-cells which abundantly express CXCR3, a G-protein coupled receptor [24]. Therefore, the selective delivery of IFNγ to the tumor site can promote an inflammatory T-cell generation and trafficking mediated by IP-10 release [22].

In addition, IFNγ can induce tumor cell apoptosis [25]. However, IFNγ may also elicit pro-tumorigenic activities [26]. For instance, IFNγ may contribute to cancer immune evasion by augmenting the expression of PD-L1 by tumor cells and dampening the anti-tumor activity of exhausted T-cells expressing PD-1 receptors [27,28]. These findings suggest that there may be a synergy between PD-1 immune checkpoint blockade (ICB) and IFNγ-based biopharmaceuticals, supporting T-cell evasion from the PD-1/PD-L1 exhaustion network. Recombinant IFNγ (ACTIMMUNE™) is approved for the treatment of infections associated with chronic granulomatous disease [29], but has also extensively been studied in cancer patients [NCT00004032, NCT00004016, NCT02614456]. However, systemic toxicity and short serum half-life have limited the systemic use of this cytokine. [30,31].

An antibody-cytokine fusion protein consisting of the L19 antibody (specific to the alternatively spliced extra domain B of Fibronectin “EDB”) in scFv format fused to a murine IFNγ variant has previously been studied in tumor-bearing mice [32]. Biodistribution studies showed that the targeting ability of the L19-IFNγ immunocytokine is influenced by the number of IFNγRs expressed in the mouse. A second immunocytokine based on the F8 antibody (specific to the alternatively spliced extra domain A of Fibronectin “EDA”) in diabody format fused to murine IFNγ did not exhibit the expected preferential localization on tumors *in vivo* indicating a potential receptor-trapping mechanism [33].

To avoid the potential receptor-trapping mechanism *in vivo* and/or to limit toxicity, IFNγ variants with reduced biological activity and receptor affinity may be considered for the targeted delivery of the payload. It has been previously described that single point mutations in the human IFNγ sequence led to a dramatic reduction in biological activity and that the carboxyl-terminal region of human IFNγ is fundamental for biological activity [34,35,36]. For example, Lundell et al. showed that certain truncated versions of IFNγ at the C-terminal region, including the 9 amino acid truncation at the C-terminal, may abrogate receptor binding and biological activity [35]. Huyghe et al. developed a truncated version of murine IFNγ fused to a VHH specific for murine CD20 [37]. The protein showed reduced biological activity in an *in vitro* cell-based assay.

The carboxy-terminal region of IFNγ is naturally highly susceptible to proteolytic digestion, suggesting that the heterogeneity of natural IFNγ may be due to differences in the degree of glycosylation and proteolytic processing of the carboxyl terminus [38].

In this work, we describe the engineering and the validation of a novel IFNγ antibody fusion protein termed L19-IFNγ KRG consisting of the L19 antibody in IgG4 format fused at the heavy chain C-terminus to a truncated version of IFNγ, which was not susceptible to proteolytic digestion at its C-terminus and showed reduced affinity to its cognate receptor and reduced biological activity. Unlike the previously described IFNγ-antibody fusion proteins, L19-IFNγ KRG selectively localized to neoplastic lesions and showed no evidence of *in vivo* trapping in PK studies performed in cynomolgus monkeys. To assess anti-cancer activity, a murine surrogate termed L19-mIFNγ KRG was generated with a similar truncation in the IFNγ moiety. Therapy experiments in immunocompetent mouse models of cancer revealed tumor growth retardation at doses that were well tolerated.

## 2. Material and Methods

### 2.1. Cell Lines

All cell lines were received between 2018 and 2021, expanded, and stored as cryopreserved aliquots in liquid nitrogen. CHO-S, CT26, F9, WEHI-164, THP-1 and TIB-49 cells were obtained from ATCC. Cells were grown according to the supplier’s protocol and kept in culture for no longer than 10 passages. DMEM (Gibco 11965) + 10% FBS (Gibco 10270106) was used for F9, THP-1 and TIB-49. F9 cells were grown on 0.1% gelatin-coated flasks (Sigma-Aldrich G1393). RPMI (Gibco 11875093) + 10% FBS (Gibco 10270106) was used for CT26, WEHI-164. Authentication of the cell lines also including checks of post-freeze viability, growth properties, and morphology test for mycoplasma contamination, isoenzyme assay, and sterility test were performed by the cell bank before shipment.

### 2.2. Cloning, Expression, and Protein Purification

The fusion protein L19-IFNγ-KRG contains the L19 antibody in IgG4 format fused to either the human IFNγ-KRG or the murine IFNγ-KRG at the C-terminus of the heavy chain. The human variant is covalently linked to the heavy chain while the murine variant is covalently linked to the heavy chain through a 15 amino acid long linker (GGGGS)_3_. The gene encoding for L19 in IgG4 format was PCR amplified while the gene encoding for the murine and human IFNγ wild-type was PCR amplified introducing an 8 AAs residues deletion and a lysine to glycine point mutation was introduced in position 153 resulting in the IFNγ-KRG mutant. The sequences encoding for the heavy chain and the IFNγ-KRG mutant were PCR assembled and cloned into pcDNA3.1 (+) by BamHI/NotI restriction sites both for the human and the murine variant. The sequence containing the light chain of the L19 antibody was PCR assembled and cloned into the same plasmid between SpeI/BsiWI restriction sites. The two fusion proteins were expressed by transient gene expression (TGE) in CHO-S cells and purified as described [39].

### 2.3. Protein Characterization

The fusion proteins described in this work were produced through TGE in CHO-S cells and purified from the cell culture medium by protein A [40] Sepharose (Sino Biological) affinity chromatography, dialyzed against phosphate-buffered saline (PBS) and stored in PBS at −80 °C. Purified proteins were analyzed by size-exclusion chromatography (SEC) using a Superdex 200 increase 10/300 GL column on an ÄKTA FPLC (Cytiva, Marlborough, MA, USA). SDS-PAGE was performed with 10% gels under reducing and non-reducing conditions. Affinity measurements were performed by SPR using a BIAcore X100 instrument (Cytiva) on CM5 EDB or IFNγR1 coated chip. Samples were injected as serial dilutions, in a concentration range from 1 µM to 7.8 nM. Regeneration of the chip was performed using NaOH 10 mM for EDB or HCl 10 mM for IFNγR1. Differential scanning fluorimetry was performed on an Applied Biosystems StepOnePlus RT-PCR instrument. Protein samples were diluted at 2 µM in PBS in 40 µL and placed in MicroAmp^®^ tubes (Applied Biosystems 4358293 + 4323032); the assay was performed in triplicates. 5× SYPRO ORANGE (Invitrogen, stock 5000×) was added to samples prior to analysis. For thermal stability measurements, the temperature range spanned from 25 °C to 95 °C with a scan rate of 1 °C/minute. Data analysis was performed in Protein Thermal Shift Software version 1.3 (Thermo Fisher) by calculating the derivative of the melting curve.

### 2.4. C-Terminus Sequence Confirmation by Intact Mass Analysis

100 μg of each fusion protein was reduced in denaturing conditions by incubation in 50 mM tris (2-carboxyethyl)phosphine (TCEP) and 4 M guanidinium chloride for 1 h at room temperature. The samples were then desalted using C18 Micro Spin Columns (Harvard Apparatus). The eluate from C18 purification was adjusted to 48% CH_3_CN/0.2% HCOOH and then directly injected into a Q Exactive mass spectrometer (Thermo Scientific) equipped with an ESI Ion Max source (Thermo Fisher). Source parameters were set as polarity positive ion mode; resolution (FWHM at 200 m/z) 70,000; microscan 10; S-lens RF level 55; spray voltage 3.5 kV; and scan range 100–1500 m/z. By applying an in-source CID offset voltage of 80 eV fragmentation of L19-IFNg variants was induced. The resulting fragmented spectra were screened for the presence of y ions, which contain C-terminal sequence information and they were manually annotated allowing a mass error of 10 ppm.

### 2.5. Bioactivity Measurement

The biological activity of L19-IFNγ KRG was evaluated by an IP-10 release assay on THP-1 and TIB-49 for the human and murine payload, respectively. Pre-coating of the wells with 0.1 μM EDB was performed overnight at 4 °C. The following day, 100 µL of cell suspension (20–25 × 10^3^ cells/well) was incubated for 48 h at 37 °C and 5% CO_2_ with a serial dilution of the IFNγ derivatives. Cultured supernatants were analyzed by a sandwich enzyme-linked immunosorbent assay (ELISA) according to the manufacturer protocol ELISA MAX™ Deluxe Set Human CXCL10 (Biolegend, 439904) for the human variant and according to mouse CXCL10/IP-10/CRG-2 (R&D systems, DY466) for the murine variant. Absorbance was measured at 450 nm and 570 nm. Relative absorbance was converted to IP-10 (pg/mL) with the respective calibration curves.

### 2.6. Immunofluorescence Infiltrate Study

For *ex vivo* infiltrate immunofluorescence analysis, mice were injected according to the therapy schedule and euthanized 24 h after the last injection. Tumors were excised and embedded in a cryo-embedding medium (ThermoScientific, Waltham, MA, USA) and the corresponding cryostat tissue sections (8–10 μm thickness) were stained using the following primary antibodies: goat anti-CD31 (R&D Systems; AF3628), rabbit anti-Foxp3 (Invitrogen, 5H10L18; 7000914), rabbi anti-NCR1 (Abcam; ab214468), rabbit anti-CD4 (Sino Biological, Wayne, PA; 50134-R001), rabbit anti-CD8 (Sino Biological; 50389-R208) and rabbit anti-mouse CD274 (BiossUSA, Boston, MA, USA; bs1103R). Primary antibodies were detected with Donkey anti-rabbit AlexaFluor488 (Invitrogen; A11008) and Donkey anti-goat AlexaFluor594 (Invitrogen; A21209). Cell nuclei were stained with 4′,6-diamidino-2-phenylindole (DAPI) (Invitrogen; D1306). Slides were mounted with a fluorescent mounting medium (Dako Agilent, Carpinteria, CA, USA) and analyzed with a wide-field Leica TIRF microscope using the Leica LAS X Life Science Microscope Software. Quantification of tumor-infiltrating cells was made using Image J software [41].

### 2.7. Experimental Animals

A total of 97 female BALB/c and 8 129/SvEv mice, aged 8 weeks with an average weight of 20 g, were used in this work. Mice were purchased from Janvier (Route du Genest, 53,940 Le Genest-Saint-Isle, France) and raised in a pathogen-free environment with a relative humidity of 40–60%, at a temperature between 18 °C and 26 °C and with daily cycles of 12 h light/darkness according to guidelines (GV-SOLAS; FELASA). Animals were kept in groups of 5 or fewer mice per cage and were reallocated in case of single housing to another cage. Blinding of the experimental groups was not performed, animals were enrolled in experimental groups according to their tumor volume (i.e., when tumors reached a volume between 80–110 mm^3^). Mice were monitored daily; tumor volume was measured with a caliper (volume = length × width × 0.52).

### 2.8. Biodistribution Experiments

10^7^ F9 tumor cells were injected subcutaneously into the right flank of 129/SvEv mice. When tumors reached a volume of 100 to 200 mm^3^, 100 μg of L19-mIFNγ-KRG or L19-IFNγ-KRG were labeled with ^125^I and Chloramine T, filtered on a PD10 column and inoculated into the lateral tail vein as described [42]. Mice were euthanized 48 h after injection. Organs, blood, and tumors were weighed, and radioactivity was detected using a Packard Cobra gamma counter. The immunocytokine uptake in blood, organs, and tumors was calculated and expressed as the percentage of the injected dose per gram of tissue (%ID/g ± SEM, n = 4). Data were adjusted for tumor growth as described [43].

### 2.9. Non-Human Primate Pharmacokinetics Studies

Adult Cynomolgus monkeys estimated to weigh between 2.4 and 3.9 kg were used in this study. Test items were administered by bolus intravenous injection at a dose volume of 1.0 mL/kg body weight (corresponding to 0.1 mg/kg and 0.5 mg/kg). The dose was administered to each animal based on the body weight measured on the day of administration. Blood samples of ∼0.6 mL each were collected at the following time points: before dosing and at 2, 10, 20, 30, 60, 120, and 240 min after treatment. Samples were transferred into serum separator tubes, kept for 30 min in an upright position then centrifuged at 4 °C (2300 g for 10 min). Fusion protein concentrations in serum were assessed by AlphaLISA. Briefly, Streptavidin Donor Beads were coated with biotinylated antigen (EDB). Acceptor beads coated with an anti-IFNγ antibody were used for detection.

### 2.10. Dose Escalation Study

WEHI-164 tumor cells were injected subcutaneously in the flank of Balb/C mice using a total of 5 × 10^6^ cells. When tumors reached a volume of 100 mm^3^ mice were intravenously injected at the following dose and schedules: 400 µg, 200 µg, and 100 µg with a single injection; 60 µg, 40 µg and 20 µg two injections with 48 h intervals between injections and 20 µg, 10 µg and 5 µg three injections with 72 h interval between injections. Tumor volume and body weight was monitored daily after the first injection. Body weight loss equal or exceeding to 15% with respect to the average body weight was considered as an endpoint.

### 2.11. Toxicity Assessment

BALB/c mice were injected three times into the lateral tail vein with L19-mIFNγ KRG (20 µg). The fusion protein was dissolved in Ringerfundin, also used as a negative control, and administered every 72 h. One day after the last injections mice were sacrificed and a necropsy was performed by a veterinary pathologist. An initial macroscopic examination of the external surface of the body, all orifices, the cranial, thoracic, and abdominal cavities and their contents and organs and tissues from every animal was performed. Selected tissues were fixed in 10% neutral buffered formalin, dehydrated, embedded in paraffin, sectioned, stained with hematoxylin and eosin, and examined microscopically.

### 2.12. Subcutaneous Tumor Model

Tumor cells were implanted subcutaneously in the right flank using 5 × 10^6^ cells (CT26) or 5 × 10^6^ cells (WEHI-164). When tumors reached a suitable volume (between 80–110 mm^3^), mice were randomized and intravenously injected with 20 µg of the immunocytokine preparation dissolved in saline solution, also used as the negative control, every third day for three times. In the combination groups, mice received L19-mIFNγ-KRG as described, followed by 200 µg of an immune checkpoint inhibitor either mouse anti-PD-1 (BioXcell, clone 29F.1A12, cat BE0273), mouse anti-PD-L1 (BioXcell, clone 10F.9G2, cat BE0101) or anti-LAG-3 (BioXcell, clone C9B7W, cat BE0174) 24 h later. For the CT26 study with chemotherapeutic agents, a first intraperitoneal injection of either oxaliplatin (2.5 µg/g) or irinotecan (9 µg/g) was followed by an intravenous injection of 20 µg L19IFNγ-KRG 6 h later. Doses of chemotherapeutic agents were translated and adapted from human dosage regimens. Subcutaneous tumors presenting ulceration and/or exceeding 15 mm in length and/or width were sacrificed as an endpoint.

### 2.13. Statistical Analysis

Data were analyzed using Prism V.9.0 (GraphPad Software). Statistical significance between multiple groups was evaluated with the one-way analysis of variance (ANOVA) followed by Tukey’s post-test. Differences in tumor volume between therapeutic groups were compared using the two-way ANOVA or mixed effects analysis followed by Tukey’s post-test. *p* < 0.05 was considered statistically significant. (*: *p* < 0.05, **: *p* < 0.01, ***: *p* < 0.001, ****: *p* < 0.0001).

### 2.14. Ethical Statements

Mouse experiments were performed under a project license (license number 06/2021) granted by the Veterinäramt des Kantons Zürich, Switzerland, in compliance with the Swiss Animal Protection Act (TSchG) and the Swiss Animal Protection Ordinance (TSchV). Procedures on Cynomolgus monkeys (including housing, health monitoring, restrain, and dosing) and ethical revision were performed according to the current Italian legislation (Legislative Decree 4 March 2014, n. 26) enforcing the 2010/63/EU Directive on the protection of animals used for biomedical research.

## 3. Results

### 3.1. Production and Characterization of L19-IFNγ variants

To circumvent the potential in vivo receptor trapping observed with biopharmaceuticals bearing an IFNγ wild-type payload [32,33], three fusion proteins consisting of the L19 antibody in IgG4 format were fused at the C-terminus of the Heavy Chain to different truncated IFNγ moiety (Figure 1A). The first molecule (termed L19-IFNγ TG) featured the deletion of 11 amino acids at the C-terminus of the IFNγ moiety (Appendix A). A second molecule (termed L19-IFNγ KR), in which 9 amino acids at the C-terminus of the IFNγ moiety had been deleted, was also produced (Appendix A). Finally, a third molecule (termed L19-IFNγ KRG), in which a C-terminal glycine was added to the L19-IFNγ KR variant, was generated (Appendix A). The fusion protein (termed L19-IFNγ WT) without changes to the IFNγ structure was used as a control (Appendix A). A BIAcore analysis on a sensor chip coated with IFNgR1 γ revealed a complete loss of binding to this cognate receptor component for both L19-IFNγ TG and L19-IFNγ KR variants, compared to L19-IFNγ WT. By contrast, only a partially reduced binding was observed for L19-IFNγ KRG (Appendix A).

### 3.2. C-terminus Sequence Confirmation by Intact Mass Analysis 

Antibody-based fusion proteins are sometimes cleaved during production, yielding heterogeneous products unsuitable for pharmaceutical development. Intact mass analysis was used to characterize which among the generated IFNγ fusion protein variants had the expected C-terminus or were cleaved. For L19-IFNγ-KRG, by performing in-source CID fragmentation, an informative spectrum was obtained which revealed several y-ions comprising the expected C-terminus of the heavy chain of L19-IFNγ, confirming its identity (Appendix A). For L19-IFNγ-KR the anticipated C-terminus fragments were not detected. Instead, y-ions related to the truncated version L19-IFNγ-TG (i.e., after the loss of Arginine and Lysine) were identified (Appendix A). Based on these results the L19-IFNγ-KRG product was selected for further *in vitro* and *in vivo* characterization.

### 3.3. Generation and Biochemical Characterization of L19-IFNγ KRG

The fusion protein L19-IFNγ KRG was cloned and produced by transient gene expression in mammalian CHO cells. The immunocytokine could be produced to homogeneity, as shown by SDS-PAGE (Figure 1B) and gel filtration (Figure 1C). The binding kinetics of the L19 antibody to its cognate antigen (EDB) was confirmed by BIAcore analysis (Figure 1D). Differential scanning fluorimetry showed a first transition at 44.8 °C, which could be attributed to the IFNγ payload. Two additional transition peaks were observed at 64.8 °C and 74.3 °C for the F_ab_ and the F_c_ domains [44,45] respectively, as confirmed by the L19 IgG4 control exhibiting two peaks at the same temperatures (Figure 1E). 

### 3.4. Functional Characterization of L19-IFNγ KRG

A BIAcore analysis on a sensor chip coated with IFNγR1 was performed, in order to compare the binding kinetics of the L19-IFNγ WT payload (Figure 2A) with the ones of the L19-IFNγ KRG truncated variant (Figure 2B). The biological activity of L19-IFNγ KRG was evaluated by an IP-10 release on THP-1 cells in the presence or absence of the target antigen EDB. L19-IFNγ WT did not exhibit substantial changes in IP-10 release in the presence or absence of EDB coating (Figure 2C). By contrast, L19-IFNγ KRG showed reduced IP-10 levels, which were restored upon binding to the cognate EDB antigen (Figure 2D). IP-10 levels released by exposure to recombinant IFNγ are shown in Figure 2E. 

### 3.5. Pharmacokinetic Profiles of L19-IFNγ KRG and L19-IFNγ WT in Monkeys

Pharmacokinetics of L19-IFNγ KRG and L19-IFNγ WT were evaluated in Cynomolgus monkeys injected once at a dose of 0.1 mg/kg and 0.5 mg/kg (only for L19-IFNγ KRG). Blood samples were collected at 2, 10, 20, 30 min and 1, 2, 4 h after injection. At 0.1 mg/kg L19-IFNγ KRG revealed a superior profile compared to L19-IFNγ WT, without revealing any in vivo trapping events at early time points (Figure 2F). Additionally, no evidence of *in vivo* trapping was observed for L19-IFNγ KRG at 0.5 mg/kg (Figure 2G).

### 3.6. Generation and Characterization of the L19-mIFNγ KRG

Since human IFNγ does not cross-react with the murine IFNγR1 [46] (Appendix A), we generated a murine surrogate termed L19-mIFNγ KRG, featuring a similar truncation in the C-terminus of murine IFNγ and a C-terminal Glycine (Appendix A) to perform *in vivo* therapy experiments in immunocompetent mice. Figure 3A shows a schematic representation of the murine analog used for therapy experiments. As for the human variant, the product could be purified to homogeneity, as shown by SDS-PAGE (Figure 3B) and gel filtration analysis (Figure 3C). The biological activity of L19-mIFNγ KRG was evaluated by a mouse IP-10 release on TIB-49 cells in the presence or absence of the EDB target antigen (Figure 3D). Another experiment was also performed with the murine wildtype variant (Appendix A). Similar to the human payload, L19-mIFNγ KRG showed reduced IP-10 release, which was significantly increased upon binding to EDB coated on a solid support. 

### 3.7. In Vivo Quantitative Biodistribution Profile of L19 IFNγ KRG

The in vivo tumor targeting performance of L19-IFNγ KRG (Figure 4A) and L19-mIFNγ KRG (Figure 4B) was assessed by quantitative biodistribution experiments in F9 tumor-bearing mice using radiolabeled protein preparations. Both the human and murine variants preferentially localized to neoplastic lesions with a tumor:blood ratio of 6 and 5, respectively, 48 h after intravenous administration. The higher uptake in the tumor of the fully human product could be attributed to the lack of cross-reactivity of the human cytokine payload to the murine receptor.

### 3.8. Dose Escalation Study and Histopathological Analysis

A dose escalation with L19-mIFNγ KRG was performed on tumor-bearing Balb/c mice to determine the optimal dose and schedule. In the first experiment, a single bolus injection of either 400 μg, 200 μg, and 100 μg was tested, but toxicity was observed 72 h after intravenous administration (Appendix A). In a second experiment, three i.v injections were performed at doses of 60 μg, 40 μg and 20 μg every 48 h. Toxicity was observed after 4 days for the 60 μg and 40 μg groups and at day 5 for the 20 μg group. (Appendix A). In a third experiment, injections were performed at 72 h intervals at the following doses: 20 μg, 10 μg and 5 μg per mouse. All doses were tolerated and showed no sign of toxicity (Appendix A). Based on these results, the 20 µg dose per mouse (administered three times every 72 h) was selected for further studies. Histopathological analysis was performed on tumor-bearing mice treated at the recommended dose and compared to mice treated with saline solution. Both treated and non-treated animals showed a mildly enlarged spleen due to lymphoid hyperplasia and extramedullary hematopoiesis, which are frequently found in experimental mice, and thus not related to the treatment. No abnormality was observed in the rest of the organs. (Appendix A).

### 3.9. Therapy Experiments

In the first experiment, we compared the therapeutic activity of L19-mIFNγ KRG, KSF-mIFNγ KRG (the KSF antibody, being specific to hen egg lysozyme, was used as a negative control of irrelevant specificity in the mouse; Appendix A), and a commercial anti-mouse PD-1 antibody in immunocompetent Balb/c mice, bearing CT26 colorectal carcinomas (Figure 5A). Moreover, L19-mIFNγ KRG was combined with anti-PD-1. In this setting, L19-mIFNγ KRG induced tumor growth retardation compared to mice treated with saline or with KSF-mIFNγ KRG. Mice treated with anti-PD-1 showed a similar growth retardation profile as in L19-mIFNγ KRG treated mice. The combination of the two agents was marginally superior. All treatments were well tolerated.

In a second therapy experiment, we combined L19-mIFNγ KRG with a commercial anti-LAG3 antibody (Figure 5B) in the CT26 tumor model. Also in this setting, the combination treatment induced tumor growth retardation compared to saline treatment (not statistically significant). Treatments were tolerated as evidenced by body weight profiles.

In a third therapy experiment, L19-mIFNγ KRG was combined with a commercial anti-PD-L1 (Figure 5C), oxaliplatin (Figure 5D) or irinotecan (Figure 5E) in the CT26 model. The combination of L19-mIFNγ KRG with anti-PD-L1 was not superior compared to single agents. Similar findings were observed in the combination with oxaliplatin, whereas an increased tumor growth retardation was found in mice treated with the L19-mIFNγ KRG + irinotecan combination. Also in this therapy experiment, all treatments were tolerated.

Based on the encouraging results obtained in the CT26 model in combination with anti-PD-1, an additional tumor therapy experiment was performed in the WEHI-164 sarcoma model (Figure 5F). L19-mIFNγ KRG confirmed its ability to induce tumor growth retardation compared to saline and KSF-mIFNγ KRG treatments. Treatment with anti-PD-1 was able to induce 2 out of 6 complete remissions, while the combination of L19-mIFNγ KRG with anti-PD-1 induced 3 out of 6 complete responses.

### 3.10. Microscopic Analysis and Quantification of Tumor Infiltrating Lymphocytes

A microscopic analysis of tumor sections obtained 24 h after the last injection of saline, L19-mIFNγ KRG, anti-PD-1 or the combination revealed a substantial increase in the density of CD4+ T cells, CD8+ T cells and NK cells only in mice treated with the combination L19-mIFNγ KRG + anti-PD-1 in both CT26 (Figure 6) and WEHI-164 (Figure 7) tumor models. By contrast, PD-L1 expression levels were similar in all specimens and Tregs were not detected in the tumor mass.

## 4. Discussion

In this work, we described the generation of a new IFNγ-based immunocytokine named L19-IFNγ KRG. The product was engineered at the C-terminus of the IFNγ moiety to have reduced affinity to its cognate receptor with the aim of preventing potential *in vivo* trapping mechanisms described before for other IFNγ-based products [32,33]. The new prototype showed reduced biological activity *in vitro* which was regained upon EDB binding by means of the L19 antibody. L19-IFNγ KRG showed excellent *in vivo* tumor-targeting properties and modest tumor growth retardation, which was enhanced when the product was used in combination with immune checkpoint inhibitors or with standard chemotherapy. The PK profile in monkeys was similar to the one reported for other IgG-based products [47]. It would be attractive to use a long-lived product (such as IgGs) to reduce dosing frequency in patients. Moreover, the truncated IFNγ payload would be inactive in blood, thus reducing systemic toxicity, and would regain biological activity at the site of disease upon antigen binding.

IFNγ is a pleiotropic cytokine that can boost anti-tumorigenic activities through various mechanisms [48,49], but at the same time, can promote cancer cell growth [50,51]. Several groups, including ours, have previously reported the curative potential of IL12-based pharmaceuticals in pre-clinical experiments. Tumor-targeted IL12 products typically led to a significant elevation of IFNγ levels, both in the tumor and in the blood [52,53]. Similar observations were reported also in cancer patients [54]. In view of these findings, IFNγ is considered the key mediator of the anti-cancer activity triggered by IL12. It was thus attractive to generate a product that can selectively deliver IFNγ at the site of disease.

Song and colleagues [55] postulated that the effect of IFNγ on tumor cells is strictly dose-dependent. In their study, low doses of IFNγ appeared to favor tumor growth, while high doses could eradicate NSCLC xenografts [55]. A possible explanation of this phenomenon relies upon an equilibrium between two possible signaling cascades IFNγ may trigger upon binding to its receptor. Similar findings have been reported for other cytokines. For instance, a high dose of IL2 is approved for the treatment of melanoma and renal cell carcinoma, while the same payload given at a low dose is used for treating chronic inflammatory conditions [56].

Additionally, another study has reported that tumor cell stemness might cause PD-1/PD-L1 resistance [57]. IFNγ may increase the PD-L1 expression on tumor cells [27,58], thus the rationale of our work was to combine L19-IFNγ KRG with anti-PD1 and anti-PD-L1 antibodies.

Our work showed that the administration of a targeted attenuated version of IFNγ can trigger the host immune response leading to tumor growth retardation. A deeper understanding of IFNγ biology may be beneficial to generate novel engineered prototypes with improved curative anti-cancer potential.

## Figures and Tables

**Figure 1 pharmaceutics-15-00377-f001:**
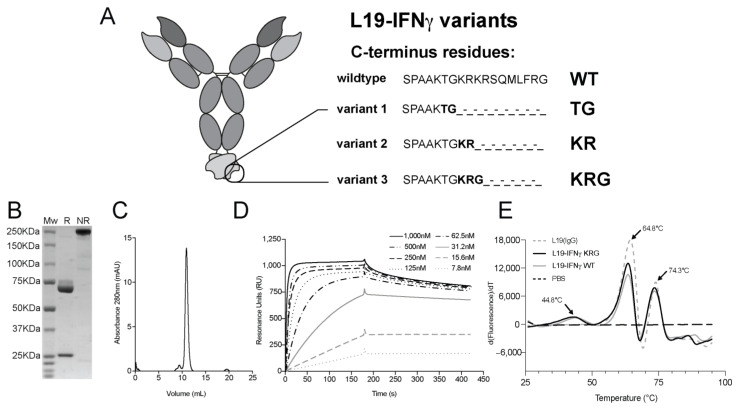
L19-IFNγ variants and biochemical characterization of fully human L19-IFNγ KRG. (**A**) Schematic representation of L19-IFNγ variants and relative C-terminus residues according to truncation. (**B**) SDS-Page analysis on 10% gel in reducing (R) and non-reducing conditions (NR) of L19-IFNγ-KRG. (**C**) Size exclusion chromatogram of L19-IFNγ KRG. (**D**) SPR of L19-IFNγ-KRG on EDB-coated CM5 sensor chip. (**E**) Differential scanning fluorimetry of L19 IgG, L19-IFNγ KRG and L19-IFNγ WT.

**Figure 2 pharmaceutics-15-00377-f002:**
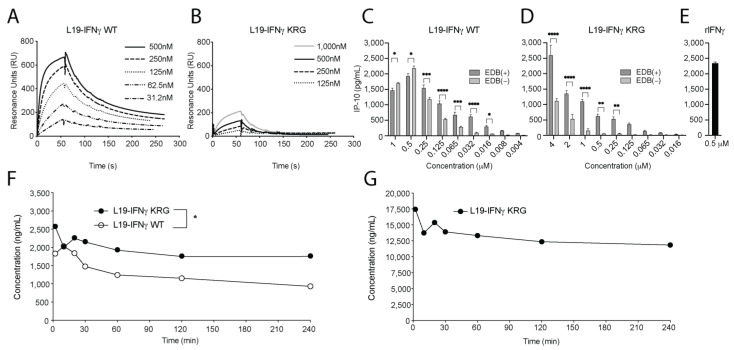
Fully human L19-IFNγ KRG functional characterization. (**A**) SPR of L19-IFNγ WT on human IFNγR1. (**B**) SPR of L19-IFNγ KRG on human IFNγR1. (**C**) IP-10 release on THB-1 cells exposed to titration of L19-IFNγ WT in coated EDB (+) and non-coated EDB (−) wells. (**D**) IP-10 release on THB-1 cells exposed to titration of L19-IFNγ KRG in coated EDB (+) and non-coated EDB (−) wells. (**E**) IP-10 release on THB-1 cells exposed to recombinant IFNγ. (**F**) Pharmacokinetic analysis conducted in cynomolgus monkeys injected at the dose of 0.1 mg/kg of L19-IFNγ KRG and L19-IFNγ WT. (**G**) Pharmacokinetic analysis conducted in cynomolgus monkeys injected at the dose of 0.5 mg/kg of L19-IFNγ KRG. (*: *p* < 0.05, **: *p* < 0.01, ***: *p* < 0.001, ****: *p* < 0.0001).

**Figure 3 pharmaceutics-15-00377-f003:**
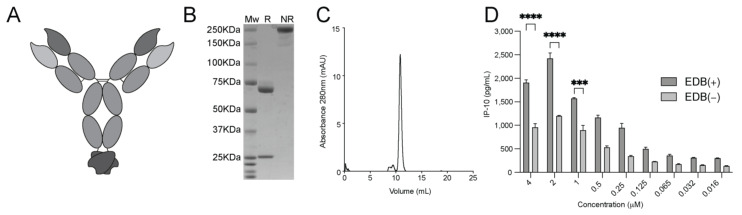
Murine L19-IFNγ KRG characterization. (**A**) Schematic representation of L19-mIFNγ KRG. (**B**) SDS gel electrophoresis, 10% gel in reducing (R) and non-reducing conditions (NR) of L19-mIFNγ-KRG. (**C**) Size exclusion chromatogram of L19-mIFNγ-KRG. (**D**) IP-10 release on TIB-49 murine leukaemia cells exposed to titration of L19-mIFNγ KRG in coated EDB (+) and non-coated EDB (−) wells. (***: *p* < 0.001, ****: *p* < 0.0001).

**Figure 4 pharmaceutics-15-00377-f004:**
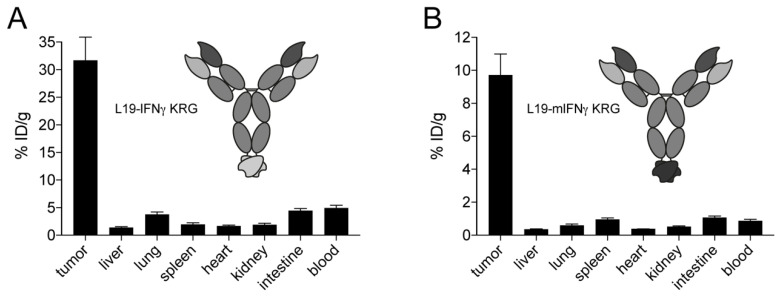
Tumor homing properties of L19-IFNγ KRG and L19-mIFNγ KRG. Quantitative biodistribution analysis of radioiodinated L19-IFNγ KRG (**A**) and L19-mIFNγ-KRG (**B**) in immunocompetent mice bearing F9 teratocarcinoma tumors. 10 µg of radio-labelled fusion protein was injected into the lateral tail vein and mice were sacrificed 48 h after injection, organs and tumor were excised, weighed and the radioactivity of each sample was measured. Results are corrected on tumor growth and expressed as percentage of injected dose per gram of tissue (%ID/g ± SEM), (n = 4 mice per group).

**Figure 5 pharmaceutics-15-00377-f005:**
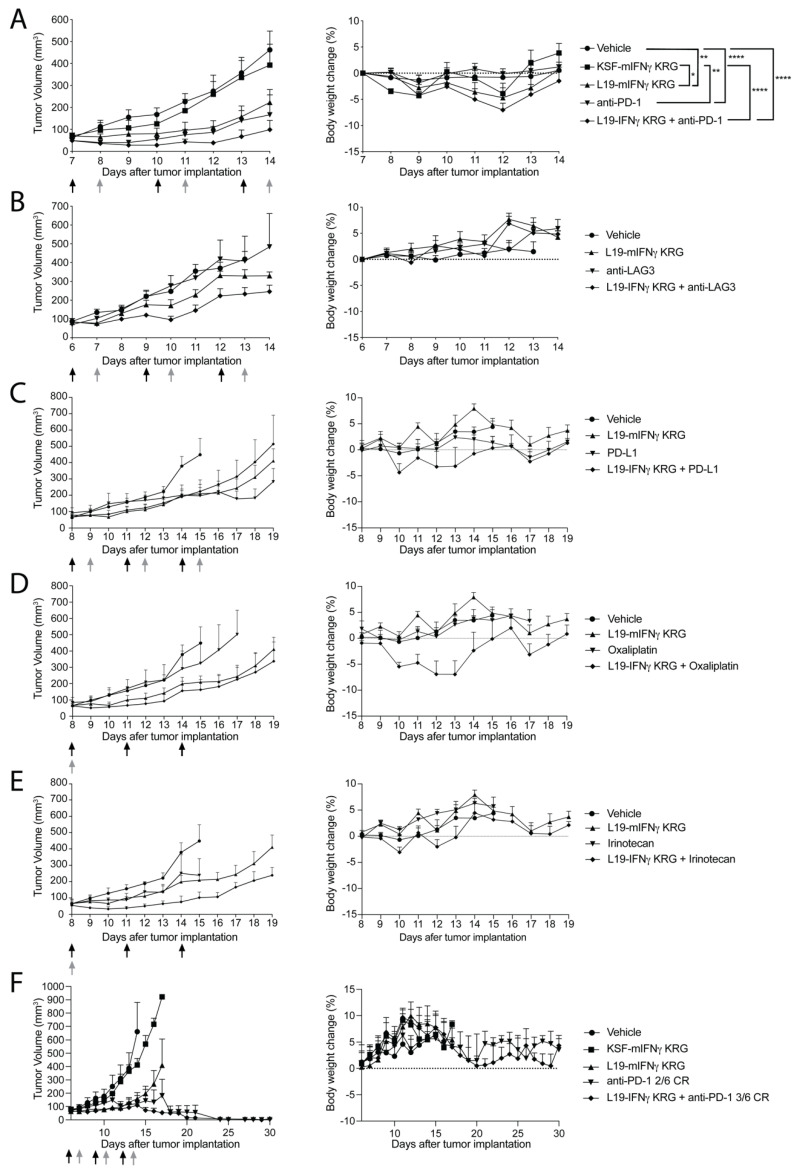
Therapeutic performance of L19-mIFNγ KRG. (**A**) CT26 tumor bearing mice received either Ringerfundin, KSF-mIFNγ KRG 20 μg, L19-mIFNγ KRG 20 μg, anti-PD-1 200μg (black arrows) three times every 72 h or L19-mIFNγ KRG 20 μg in combination with anti-PD-1 200 μg (grey arrows; n = 5 mice per group). (**B**) CT26 tumor bearing mice received either Ringerfundin, anti-LAG-3 200 μg, L19-mIFNγ KRG 20 μg, (black arrows) three times every 72 h alone or in combination with anti-LAG-3 200 μg (grey arrows; n = 3 mice per group). Third Therapy in CT26 tumor bearing mice treated with either Ringerfundin, L19-mIFNγ KRG 20 μg, anti-PD-L1 200 μg (**C**), oxaliplatin 2.5 µg/g (**D**) or irinotecan (9 µg/g) (**E**) or the combination administered three times. (n = 3–4 mice per group) Irinotecan alone or in combination received one single i.p. injection (grey arrow). (**F**) Therapy study in BALB/c mice bearing WEHI-164 lesions. The same dose and schedule treatment of (**A**) was used. Data represent mean tumor volume and body weight change % (±SEM). CR = Complete Response. (*: *p* < 0.05, **: *p* < 0.01, ****: *p* < 0.0001).

**Figure 6 pharmaceutics-15-00377-f006:**
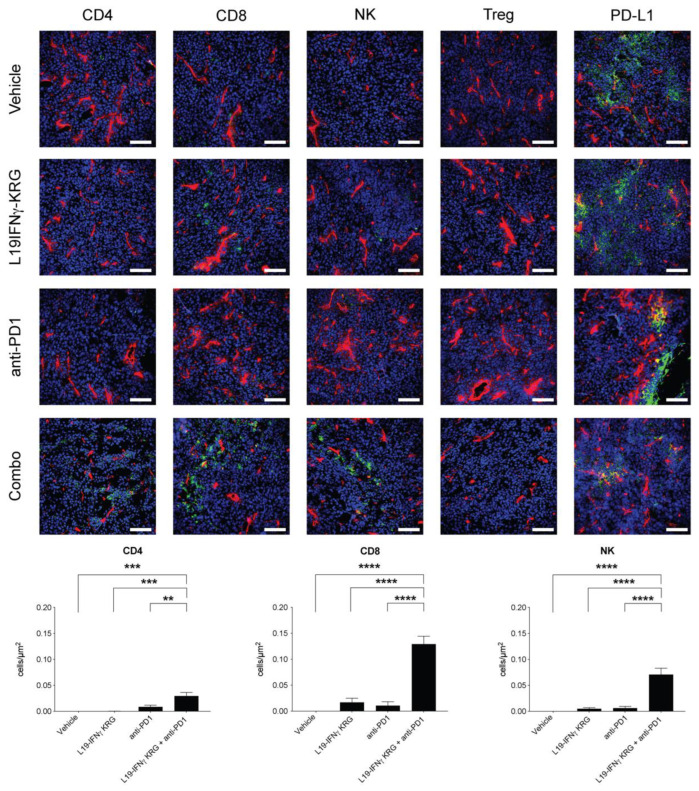
Microscopic analysis and quantification of tumor-infiltrating cells in CT26 tumor sections. Ex vivo immunofluorescence analysis on CT26 colon carcinoma 24 h after the third injection of Vehicle, anti-PD-1 and L19-mIFNγ KRG alone or in combination with 200 μg of anti-PD-1. Markers specific for, CD4 + Tcells (CD4), CD8 + Tcells (CD8), NK cells (NCR1), Tregs (Foxp3), and PD-L1 were used (green). Blood vessels were stained with an anti-CD31 antibody (red). Magnification:10×; scale bars = 100 μm. The panel below shows the quantification of CD4+, CD8+ and NK cells through imaging software. Three distinct slide sections were used for each group and cell subset. Results are expressed as number of cells and normalized on the section area (±SEM). (**: *p* < 0.01, ***: *p* < 0.001, ****: *p* < 0.0001).

**Figure 7 pharmaceutics-15-00377-f007:**
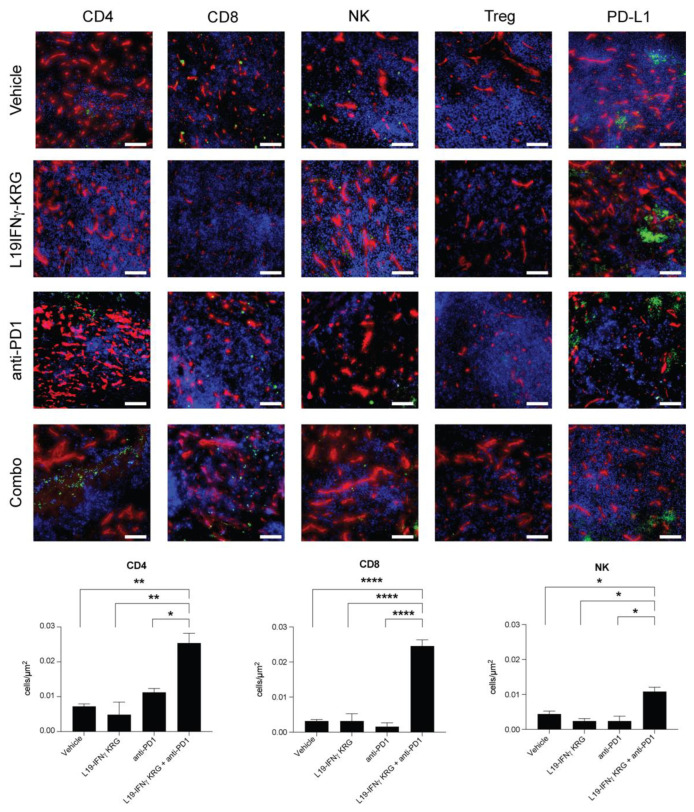
Microscopic analysis and quantification of tumor-infiltrating cells in WEHI-164 tumor sections. Ex-vivo immunofluorescence analysis on WEHI-164 sarcoma 24 h after the third injection of Vehicle, anti-PD-1 and L19-mIFNγ KRG alone or in combination with 200 μg of anti-PD-1. Marker specific for, CD4 + Tcells (CD4), CD8 + Tcells (CD8), NK cells (NCR1), Tregs (Foxp3), and PD-L1 were used (green). Blood vessels were stained with an anti-CD31 antibody (red). Magnification:10×; scale bars = 100 μm. The panel below shows the quantification of CD4+, CD8+ and NK cells through imaging software. Three distinct slide sections were used for each group and cell subset. Results are expressed as number of cells and normalized on the section area (±SEM). (*: *p* < 0.05, **: *p* < 0.01, ****: *p* < 0.0001).

## Data Availability

All data relevant to the study are included in the article or uploaded as Appendix A.

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
