# Peer review of "An Engineered IFNγ-Antibody Fusion Protein with Improved Tumor-Homing Properties"

_pharmaceutics, 2023, doi:10.3390/pharmaceutics15020377_

Round 1

Reviewer 1 Report

The authors have developed a novel fusion protein, L19-IFNg KRG, made up from an IFNg variant with a reduced affinity for its receptor and the L19 antibody, and tested it in several tumor models, both on its own and in combinations with various known anti-tumor pharmaceuticals. The results of this study show that L19-IFNg KRG preferentially homes to the tumor and limits tumor growth.

I have several comments/questions:

- In the introduction, the nature and significance of IP-10 should be explained in more detail.

- At line 50, “Upon binding to its receptor (expressed on CD4+, CD8+T-cells and NK-cells), IFNg can trigger biologically relevant anti-tumor activity by (i) boosting the expression of MHC class I molecules on tumor cells [21], (ii) recruiting T-cells at the site of disease through IP-10 release [22], and (iii) trigger tumor cell apoptosis [23]”.

The sentence is overly condensed and should be rephrased for clarity.

- The creators of Image J should be cited for the use of their program; you can find the instructions here https://imagej.net/contribute/citing

- At line 302, “At 0.1 mg/kg L19-IFNg KRG revealed a superior profile compared to L19-IFNg WT...” and “Similar findings were obtained for L19-IFNg KRG at 0.5 mg/kg”.

The claim about the similarity of the findings implies a superior profile of L19-IFNg KRG compared to L19-IFNg WT at 0.5 mg/kg, yet the L19-IFNg 569 WT control in Figure 2G is absent?

- At line 313, “The biological activity of L19-313 mIFNg KRG was evaluated by a mouse IP-10 release on TIB-49 cells in the presence or absence of the EDB target antigen (Figure 3D). The same experiment was performed with the murine wildtype variant (Supplementary Figure 9)” and “L19-mIFNg KRG showed reduced IP-10 release, which was restored upon binding to EDB”.

If I understand correctly, IP-10 release induced by L19-mIFNg KRG is being compared to that induced by L19-IFNg WT. If that is the case, “restored IP-10 release” would not be a fitting description of the obtained results, as the IP-10 concentrations in Supplementary Figure 9 are still significantly higher than those achieved in Figure 3D. Additionally, you shouldn’t call it “the same experiment”, as the highest titrated concentrations of L19-313 mIFNg KRG (2 and 4 uM) have not been tested for L19-mIFNg WT in Supplementary Figure 9.

- At line 354, “In a second therapy experiment, we combined L19-mIFNg KRG with a commercial anti-LAG3 antibody (Figure 5B) in the CT26 tumor model. Also in this setting, the combination treatment induced tumor growth retardation compared to saline treatment.”

If statistical analysis doesn’t confirm this statement, please indicate that the difference between groups in tumor growth retardation, even though noticeable, wasn’t statistically significant.

- Figure 5 shows the progression of tumor growth in various settings. The tumor sizes within the vehicle groups seem to have been measured until the 14th (A), 13th (B,F), or 15th (C,D,E) day after tumor implantation. Please elaborate in the text on the choice of the cut-off tumor size, if that was the cause for stopping the further observation of tumor progression (as the graphs suggest), and if it wasn’t, provide other reasonings. Finally, exchange the current Figure 5 for a higher quality version. Figures 5D,E,F seem especially hard to read.

Reviewer 2 Report

The current manuscript described the design and evaluation of an engineered IFNr-antibody fusion protein with improved tumor-homing properties. The paper is generally well written and the conclusion is sound and supported by the data. I only a few questions:

1, the authors have applied intact mass to characterize the fusion protein, but only reported some C-terminal sequence confirmation. The intact ms of the whole molecule is not given, why?

2, why in the pharmacokinetic study, only up to 240 min measurement but not longer time point is given?

3, why only detect the immunoinfiltrate at 24 h not other longer time point? Please justify. 

Reviewer 3 Report

This is an interesting manuscript that demonstrates linking a modified version of IFNg to an antibody results in better targeting and activation of IFNg to the tumor microenvironment.  The approach is novel and demonstrates how this cytokine may be more effectively administered in the clinical environment.  I do have a few questions:

1. Is the new construct immunogenic?

2, The authors use IP-10 expression as a surrogate for IFNg activity.  Would similar results be seen if pSTAT1 was analyzed either by FC or Western blot?

3. Using the different combinations , were there increases in tumor infiltrating macrophages?  One might well expect an M1 like phenotype.

4. In animals where tumors regressed, were they rechallenged?

5. In the primate experiments, were there any changes in the composition of the PBMC?

Round 2

Reviewer 1 Report

I want to thank the authors for answering my questions and accepting my suggestions. As a final minor comment, I just want to point out that in sections 3.4 and 3.5, where Figures 1D-F are referenced, the text should reference Figures 2D-F.

Author Response

We thank the reviewer for the comment and for spotting the error. We have updated the figure's numbering in sections 3.4 and 3.5.